

# DeepFND: an ensemble-based deep learning approach for the optimization and improvement of fake news detection in digital platform

Venkatachalam K[1], Badriyya B. Al-onazi[2], Vladimir Simic[3,4], Erfan Babaee Tirkolaee[5,6] and Chiranjibe Jana[7]

[1] Department of Applied Cybernetics, University of Hradec Králové, Hradec Kralove, Czech Republic
[2] Department of Language Preparation, Princess Nourah Bint Abdulrahman University, Riyadh, Saudi Arabia
[3] Faculty of Transport and Traffic Engineering, University of Belgrade, Belgrade, Serbia
[4] Department of Industrial Engineering and Management, Yuan Ze University, Taoyuan City, Taiwan
[5] Department of Industrial Engineering, Istinye University, Istanbul, Turkey
[6] MEU Research Unit, Middle East University, Amman, Jordan
[7] Department of Applied Mathematics with Oceanology and Computer Programming, Vidyasagar University, Midnapore, India

Corresponding author
Venkatachalam K,
venkatme83@gmail.com

## ABSTRACT

Early identification of false news is now essential to save lives from the dangers posed by its spread. People keep sharing false information even after it has been debunked. Those responsible for spreading misleading information in the first place should face the consequences, not the victims of their actions. Understanding how misinformation travels and how to stop it is an absolute need for society and government. Consequently, the necessity to identify false news from genuine stories has emerged with the rise of these social media platforms. One of the tough issues of conventional methodologies is identifying false news. In recent years, neural network models' performance has surpassed that of classic machine learning approaches because of their superior feature extraction. This research presents Deep learning-based Fake News Detection (DeepFND). This technique has Visual Geometry Group 19 (VGG-19) and Bidirectional Long Short Term Memory (Bi-LSTM) ensemble models for identifying misinformation spread through social media. This system uses an ensemble deep learning (DL) strategy to extract characteristics from the article's text and photos. The joint feature extractor and the attention modules are used with an ensemble approach, including pre-training and fine-tuning phases. In this article, we utilized a unique customized loss function. In this research, we look at methods for detecting bogus news on the internet without human intervention. We used the Weibo, liar, PHEME, fake and real news, and Buzzfeed datasets to analyze fake and real news. Multiple methods for identifying fake news are compared and contrasted. Precision procedures have been used to calculate the proposed model's output. The model's 99.88% accuracy is better than expected.

Subjects Algorithms and Analysis of Algorithms, Artificial Intelligence, Data Mining and Machine Learning, Text Mining
Keywords Fake news, DeepFND, Deep learning, Ensemble model, Joint feature extraction

## INTRODUCTION

Providing multimedia content for the news through social networks is advantageous because it is cost-effective, enables easy access, and accelerates transmission. Due to these advantages, many individuals obtain their news from these sources. As a result of the rapid expansion of social networks, many platforms within the realm of social media have evolved into excellent channels for disseminating news. People increasingly rely on social media platforms to search for and consume news due to their convenience. However, this convenience also makes it easier for false information to spread and proliferate rapidly, (*Helmstetter & Paulheim, 2021*; *Zakharchenko et al., 2021*) devastatingly affecting individuals and society. People can share and forward tweets on microblogs like Twitter and Weibo, two of the most widespread online platforms. Tweets that include text and photos are more likely to draw attention than tweets containing text.

Unfortunately, creators of deceptive posts often exploit these characteristics to expedite news spread. The rapid dissemination of false information carries the potential for harmful effects on society and even the possibility of influencing the outcomes of significant public events. The early identification of false news on social media has recently become a highly active area of interest, capturing the attention of many people.

Microblogs frequently disseminate fabricated news stories. If these tweets are not verified, they can seriously damage the reputation of the microblogging platform, highlighting the critical importance of verification. Consequently, distinguishing between original and fraudulent news when reading microblogs is of the utmost significance.

Fake news recognition aims to determine whether each given post contains fake news or not. This task is often framed as a binary classification. While other sources, such as user comments on the article and reposts, can be helpful, the information gleaned from these sources in the early stages is often noisy and incomplete. Therefore, this research's primary focus is on identifying false news based on its content.

The application of machine learning is the primary strategy employed in these techniques. In the substantial body of this research, having a labeled dataset of real and false news enables training a classification model based on new attributes. This model is subsequently used to predict the accuracy of a given piece of information. There are two common categories for the characteristics utilized in these methods: (1) features that depend on the content and (2) features that depend on the context. The elements derived from the text or the actual content of the news are referred to as content-based features (*Noureen & Asif, 2017*; *Reddy et al., 2020*; *Ajao, Bhowmik & Zargari, 2019*; *Dong, Victor & Qian, 2020*). On the other hand, context-based features depend on the news context (*e.g.*, the publisher, the position of other persons in the network, and the dissemination structure) to determine whether or not the news is false. These policies have generated decent outcomes (*Zhou & Zafarani, 2019*; *Shu et al., 2019*), but they frequently need information that is tough to obtain when presented with false news. They are only active when fake news negatively impacts the community. For instance, stance identification in news comments, which is one essential approach in detecting false news, is only applicable when the network users adopt a position against the news and post their thoughts about it

*Pamungkas, Basile & Patti (2019)*. These approaches use the evidence possessed by the other users in the network (*Ahmed, Traore & Saad, 2018*). Hence, they must wait until at least some network associates have confirmed the integrity of a piece of reported information.

## Motivation

The motivation behind this work arises from the rapidly changing landscape of news consumption, where social media has become the dominant platform for people to access and share information. With the rise of social media as a primary news source, both benefits and challenges are associated with how news is disseminated through these platforms.

The primary reasons for using media for analysis are as follows: Media provides a low-cost and easily accessible means to distribute news content. Multimedia formats, such as images and videos, enhance news presentation, making it engaging and attractive to users. Additionally, news spreads quickly through social networks, allowing information to reach a broad audience rapidly. This rapid dissemination can be advantageous for spreading essential and timely news. An increasing number of individuals are choosing social media as their primary news source due to its convenience and user-friendly interface.

The rapid spread of news on social media also presents challenges, such as disseminating false news. False news can quickly go viral before its accuracy is verified, impacting individuals and society while abusing multimedia content. Unfortunately, this characteristic is exploited by fake news producers to accelerate the spread of misinformation.

The research addresses the critical need for the early detection of false news on social media. Previous approaches have utilized machine learning to classify false news using either content-based or context-based features. However, these methods often require information that may take time to become available when encountering a piece of false news.

To overcome these limitations, the researchers propose a novel approach that combines deep learning and attention mechanisms. This ensemble approach involves constructing learners with a shared deep-feature extractor and utilizing attention modules to differentiate them. This method reduces training time, memory requirements, and model complexity while effectively addressing overfitting. The unique loss function, coupled with an attention mechanism, encourages learners to focus on specific aspects of incoming news, enhancing the model's efficiency.

## Contribution

Earlier ensemble fake news algorithms (*Hakak et al., 2021*; *Aslam et al., 2021*; *Mahabub, 2020*; *Huang & Chen, 2020*; *Roy et al., 2018*; *Das, Basak & Dutta, 2022*) often trained multiple deep or shallow models individually and then aggregated the outputs of these learners using ensemble techniques like voting. This approach aimed to generate false news, resulting in many trainable parameters and an expensive training process for these

models. Additionally, they faced challenges related to scalability and susceptibility to overfitting.

We have developed a unique method for detecting false news to overcome these obstacles, incorporating elements reminiscent of deep learning and attention processes. Our learners are built upon a shared deep-feature extractor, with differentiation occurring in their attention modules. Parameter sharing proves advantageous by efficiently reducing the required training time, memory demands, and the complexity of the proposed model.

We propose using ensemble deep learning models based on a shared feature extractor for false news detection. Compared to other ensemble models, our approach demands less training time and involves fewer parameter configurations, making it less prone to overfitting. By employing an attention mechanism, we formulate a distinctive loss function that compels the learners to focus on specific aspects of incoming news. This encourages each model to operate with high efficiency.

### Organization of the article

The remaining sections of the article are structured as follows: In the next section, we will discuss various practices for detecting false news, with an emphasis on multimodal content-based approaches. "Proposed Methodology" presents the suggested ensemble model and provides implementation details. "Experimental Setup" will outline the experimental setup, while "Results and Discussion" will present the findings related to the identification of false news using the proposed approach. Finally, in "Conclusion and Future Work", we offer some concluding remarks and suggestions for further study.

## RELATED WORKS

This section concisely summarizes previous work in detecting false news and multi-task learning. In addition to the text information itself, it is usual practice to employ the transmission structure of the news on social networks to identify false news. This applies to news that merely comprises texts. *Liu et al. (2019)* reported a kernel graph attention network. They provided more fine-grained fact verification based on kernel-based attention. *Zong et al. (2021)* utilized semantic role labeling to parse each phrase containing evidence and constructed relationships between arguments to create a graph structure for information detection. *Ma & Gao (2020)* and *Bian et al. (2020)* modeled the propagation of postings on the Weibo platform using tree topologies, different from the graph structure generated in the approaches described above.

There are a few scholars who have a variety of perspectives about the path that fake news research should take. They believe that it is of utmost significance to explore the interpretability of false news detection. For instance, *Shu et al. (2019)* constructed a combined attention graph to gather the top K interpretable sentences and user comments. *Fan, Han & Wu (2022)* proposed a dual-view paradigm based on individual and group cognition for verifying interpretive claims.

All efforts were focused on the identification of fake news, a goal achievable through machine-learning techniques. *Tacchini et al. (2017)* developed a model to distinguish between hoaxes and non-hoaxes in news disseminated on social network platforms like

Facebook. This model utilized two different machine-learning algorithms. Conversely, the assessment of the content relied on what individuals had liked or shared. *Conroy, Rubin & Chen (2015)* discussed two methods that could be employed in the search for false news. Both approaches were employed concurrently to expedite and enhance the reliability of fake news detection.

All efforts were focused on the identification of fake news, a goal that could be achieved using machine-learning techniques. *Tacchini et al. (2017)* developed a model to distinguish between hoaxes and genuine news in content distributed across social network platforms like Facebook. This model was constructed using two different machine-learning algorithms. Conversely, the assessment of the content relied on what individuals had liked or shared. *Conroy, Rubin & Chen (2015)* discussed two methods that could be employed to search for false news. Both approaches were employed concurrently to expedite and enhance the reliability of fake news detection.

Numerous postings, shared materials, and news content are available in audio, video, and text formats. Some authors have primarily focused on linguistic cue techniques, employing machine learning and network study methodologies as their primary application areas. Various methods have been used to categorize different types of false news, including serious reporting. In *Rubin, Chen & Conroy (2015)*, the identification of false information on social platforms relied on an in-depth assessment of their merits and drawbacks, text analytics, and multiple predictive models. This was achieved by analyzing numerous postings authored by different users.

*Ahmed, Traore & Saad (2017)* developed a model for detecting false news by applying n-gram and machine learning methodologies during the development process. They utilized several characteristics obtained through two methods and examined them in six distinct machine-learning contexts. Using term frequency-inverse document frequency for feature extraction and support vector machines as a machine learning analyzer, they achieved improved accuracy compared to other methods. *Zhou & Zafarani (2019)* created Fake Detector, an automated model for inferring the credibility of bogus news, designed to identify fake news on social network platforms. They employed a deep diffusive neural network to analyze various characteristics, including user profile information and the connections between users and the authors of false news, in order to identify typical features of news items. *Han, Karunasekera & Leckie (2020)* introduced graph neural networks (GNNs) that employ a continually learning-based strategy for detecting fake news on social media platforms. Their analysis using GNNs was capable of handling non-Euclidean data, often avoiding specific text content and relying on hidden data for implementation. *Ozbay & Alatas (2020)* presented a technique for detecting false news, involving the analysis of supervised artificial intelligence algorithms applied to social media accounts. The authors employed twenty-three different intelligent categorization strategies to leverage publicly available data.

This technique finds applications in various domains where such data structures and statistical methods are pertinent, including geospatial analysis, directional data analysis, and other fields characterized by grouped linear data patterns (*Fan, Yang & Bouguila, 2021*). The prediction method relies on visual data, such as images or videos, and

incorporates quantified features extracted from this visual information. Visual data sources may include traffic cameras, drones, or similar devices, and the quantified features can encompass measurements like vehicle speed, density, or lane occupancy (*Chen et al., 2023*, *2022a*; *Guo et al., 2022*). It's an image-processing algorithm that segments an image into multiple regions, analyzes the brightness information within the L-channel of a color space, and applies gamma transformations to rectify nonuniform brightness across various image regions (*Qi et al., 2022*; *Ni et al., 2021*).

In RF engineering, isolation refers to how signals on one path are prevented from interfering with signals in another way (*Feng et al., 2022*). The acronym "APMSA" stands for "Adversarial Perturbation Against Model Stealing Attacks," indicating that this technique is designed to thwart attempts by adversaries to steal machine learning models by introducing adversarial perturbations (*Zhang et al., 2023*; *Li et al., 2022*). The primary application of this method is to retrieve relevant content, which could be either images or text, based on a query or search request (*Zhou & Zhang, 2022*; *Yang et al., 2022*).

"Region-aware Image Captioning *via* Interaction Learning" implies that this approach focuses on generating image captions attentive to specific regions or objects within images, achieved through learning and modeling interactions between these elements (*Liu et al., 2021*; *Cao et al., 2021*). The model is purpose-built for analyzing and classifying product-related reviews, offering valuable insights to consumers and businesses (*Wu et al., 2018*; *Jiang et al., 2021*). The study utilizes trajectory data and spatiotemporal analysis to gain a deeper understanding, with potential applications in urban planning, traffic management, and related fields (*Xiao et al., 2021*; *Liao et al., 2021*).

"Taxi Fraudulent Trip Detection From Corresponding Trajectories" describes a system or method that uses trajectory data from taxi trips to identify and prevent fraudulent activities within the taxi service industry (*Ding et al., 2020*). The data analyzed originates from extensive IoT networks or systems with a substantial volume of data generated by numerous devices (*Chen et al., 2022b*; *Zhou et al., 2023*). Knowledge graph completion involves predicting additional relationships within a structured knowledge graph based on existing data, where various relationships connect entities (*Shen et al., 2020*; *Cheng et al., 2016*).

In visual question answering, multi-modal fusion combines visual information, such as images or videos, with textual data, such as questions or captions (*Lu et al., 2023b*; *Liu et al., 2023a*). Each message in this dataset is labeled with multiple categories, and the data collection and annotation process combines manual and automated methods (*Liu et al., 2023b*; *Lu et al., 2023a*).

When applied to a four-class label on news article headlines, *Abedalla, Al-Sadi & Abdullah (2019)* employed a mixture of different deep learning approaches such as long short-term memory network (LSTM), convolutional neural network (CNN), and bidirectional LSTM (Bi-LSTM). *Fan, Han & Wu (2022)* suggested an LSTM-based model to identify erroneous complaints within an environmental complaint system. *Bhattacharya et al. (2021)* developed a Bi-LSTM-based false news detection model, representing an enhanced version of the LSTM algorithm. Utilizing blockchain networks,

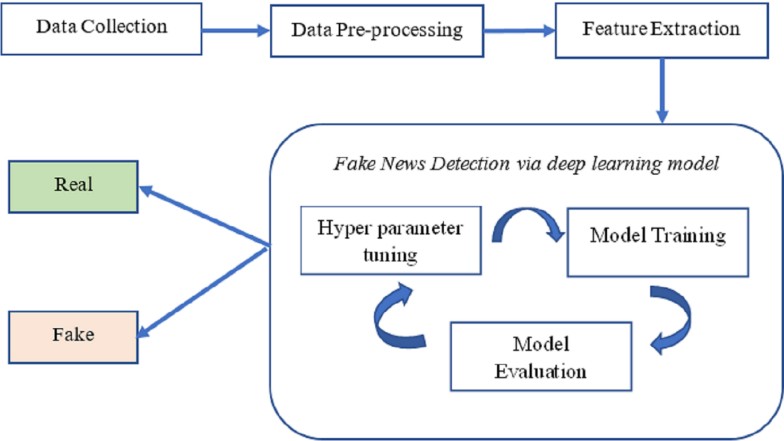

**Figure 1 The overall structure of fake news detection.**

this model offers a definitive approach to categorizing fake news and identifying news sources.

Therefore, the Bi-LSTM model could be valuable for several natural language processing applications, including phrase categorization and translation. A widely recognized CNN design, Visual Geometry Group (VGG), demonstrated that a deep network with relatively few convolutional filters could yield dependable results for the first time. The leading network incorporates an attention mechanism that assigns greater weight to the most crucial aspects. We employ a customized loss function to enhance the performance of the introduced model.

## PROPOSED METHODOLOGY

According to our methodology, the news is categorized as either "real news" or "fake news" and relies on deep neural networks. The overarching structure of the proposed system for identifying fake news is illustrated in Fig. 1. The processing pipeline for this approach comprises four steps.

Initially, we gathered news information and collected facts about fake news from various fact-checking websites. Subsequently, we cleaned the dataset to remove noise and inaccuracies, eliminating duplicate entries. The second stage, "embedding," involves using pre-trained word embeddings from GloVe to represent the data from the news articles.

In the third step, deep neural networks were trained to detect and identify fake news. These networks included Bi-LSTM and Visual Geometry Group 19 (VGG-19). The final step involves classifying and evaluating news articles (real or fake) using a testing dataset that has not been previously examined.

### The DeepFND model

The proposed model aims to evaluate the integrity of news, given textual content and an accompanying picture. Figure 2 illustrates the model's architecture, which can be divided into distinct sections.

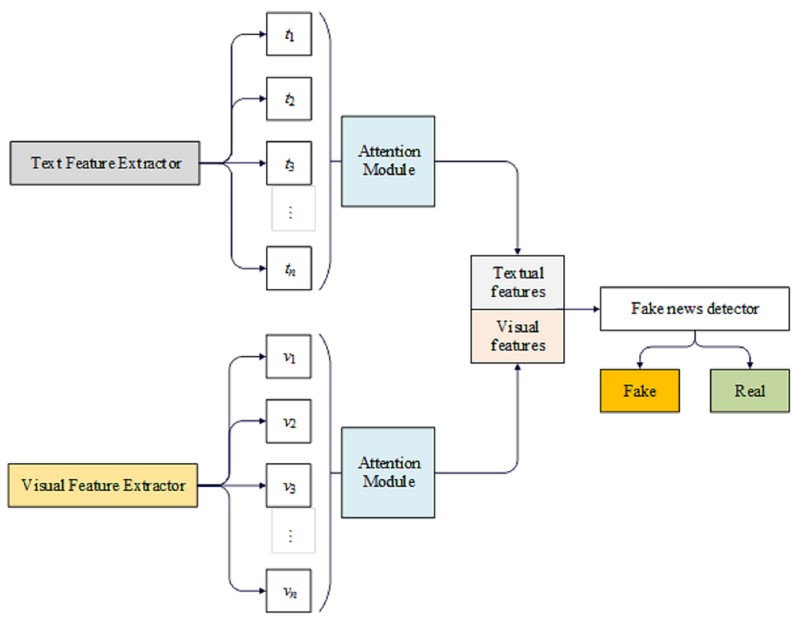

**Figure 2** The DeepFND model.               

The initial section of the system includes a textual feature extractor and a visual feature extractor, responsible for extracting textual and visual features, respectively. The subsequent step, feature fusion, employs scaled dot-product attention to create a detailed combination of textual and visual features. The final component is a fake news detector that utilizes the fused features to determine the accuracy of the news.

We have developed an innovative DeepFND model composed of four separate modules: an input module, a feature extraction module, a feature fusion module, and a detector module. Below is a comprehensive explanation of the model's underlying structure.

DeepFND is a proposed approach for identifying fake news based on the content of the story. To understand the significance of the news, we first analyze the constituent phrases. Given that various components of a post do not contribute equally to the determination of its veracity, we employ a technique that automatically learns the significance weights associated with these components. The model being presented comprises a group of learners that adhere to a common structure, simplifying the model and preventing it from becoming overly complex.

The only distinguishing factor among them is the attention modules they employ. We aim for each one to consider various aspects of the news because we recognize that the effectiveness of an ensemble model is directly related to the diversity of its learners. Therefore, in the proposed loss function, we strive to make them as dissimilar as possible. Figure 2 illustrates the architecture of the proposed model, which employs two concurrent components to extract information from both the picture and the text of a given news article. Subsequently, the extracted features are combined and categorized. In the following section, further discussion will be provided on various aspects of the model.

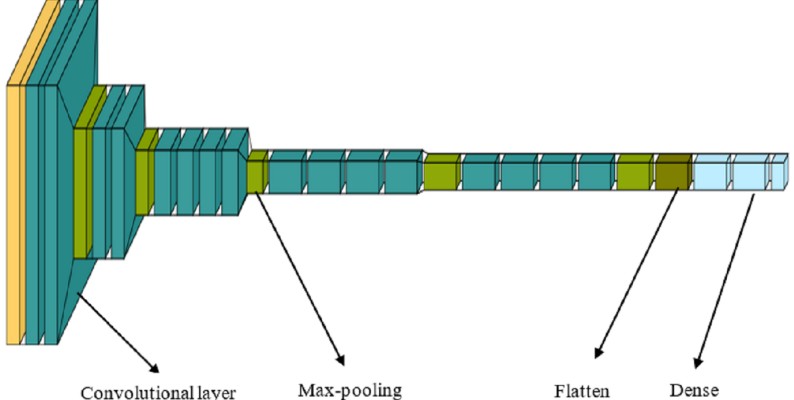

**Figure 3 VGG-19 layer architecture.**

## Visual feature extraction

VGG-19 and Bi-LSTM serve as the foundational components of our ensemble model technique. A statement often presents its data linearly, and we employ the BiLSTM architecture to capture such sequential data. Bi-LSTM is renowned for its capability to record data in both forward and backward directions.

Since even a human expert may require assistance in distinguishing real news from fake news, manually identifying appropriate features and distinguishing genuine from fake news is a technically challenging task, particularly in binary classification scenarios. It is widely recognized that VGG-19 excels at capturing hidden features. Our working hypothesis is that VGG-19 will be able to recognize latent elements within the provided statement and information related to the claims, thereby enabling an evaluation of the credibility of each claim.

The application of CNNs has yielded significant advancements in computer vision. CNNs generate multiple feature maps, which can be considered as visual characteristics of an image, by performing convolutional operations with various convolution kernels on an input image. Rather than employing a single graphical representation to represent the image, we use multiple visual features, each represented by a feature vector, and seamlessly integrate them with textual information. VGG-19, comprising 16 convolutional layers and three feed-forward layers, is utilized to learn various attributes of the images. Unlike some other networks, VGG-19 generates only a single feature vector for each image, making it challenging to fuse these data with text at a finer granularity level. Consequently, the last three fully connected layers of VGG-19 are removed, and several additional convolutional layers are inserted after the 16 convolutional layers of VGG-19 (Fig. 3).

Thus, the visual feature extractor is made up entirely of convolutional layers and produces a fixed number of feature maps:

$$f = [f_1, f_2, f_3, \ldots, f_k], \tag{1}$$

where $k$ is set by the number of convolution kernels in the final convolutional layer, and each feature map $f_i$ is a vector with dimensions (*height $\times$ width*). The visual features are:

$$vf = [vf_1, vf_2, vf_3, \ldots, vf_k], \tag{2}$$

where each feature is a single *height $\times$ width $\times$ 1* dimensional vector obtained by compressing the spatial dimensions of each feature map $f_i$ ($i = 1, 2, \ldots, k$).

Figure 4 illustrates the Bi-LSTM architecture. VGG-19 and Bi-LSTM yield superior results when their respective representations are combined. Each dense network that follows the Bi-LSTM networks undergoes reconfiguration and is subsequently passed to additional convolutional layers. In these layers, they are equipped with fresh insights about the statement, the speaker's occupation, and the surrounding context. Following each convolution layer is a max-pooling layer, which compresses the data before it is fed into their dense layers. The thick layers of multiple networks, each carrying distinct attribute information, are paired and used to uncover relationships among the various attributes. The resulting network is then fed into a dense layer with six neurons, employing softmax as the activation function. Adadelta is the optimization method, and the loss function used is categorical cross-entropy.

*Algorithm: Fake and Real news detection using VGG-16 and Bi-LSTM model*

*Input: Collected dataset $d = \{n_1, n_2, n_3, \ldots, n_k\}$*

*Output: Real and fake news classification*

*For each i in the dataset d*

  *$V_i = \{v_2, v_3, \ldots, v_k\}$*

*Apply attention mechanism*

  *$T_i = \{t_2, t_3, \ldots, t_k\}$*

*Apply attention mechanism*

*Concatenate $V_i$ and $T_i$ as $f_i$*

*Pass input $f_i$ to the ensemble model*

*Prediction (feature list)*

*If predict==1*

*Result Fake*

*Else*

*Result Real*

*Classified result of fake and real news*

*Analyze the performance based on the classification*

## Text feature extractor

Essentially, a phrase is just a string of words. Let us say $u_{kl}$ is the $l$th word in the $k$th phrase, as determined by a word embedding. This permits us to represent a sentence as $u_{k1}, u_{k2}, \ldots, u_{kl_n}$, where $l_n$ is the total number of words in the phrase. The sentence encoding process should convert this string into a $N_n$ vector of constant length. It may be

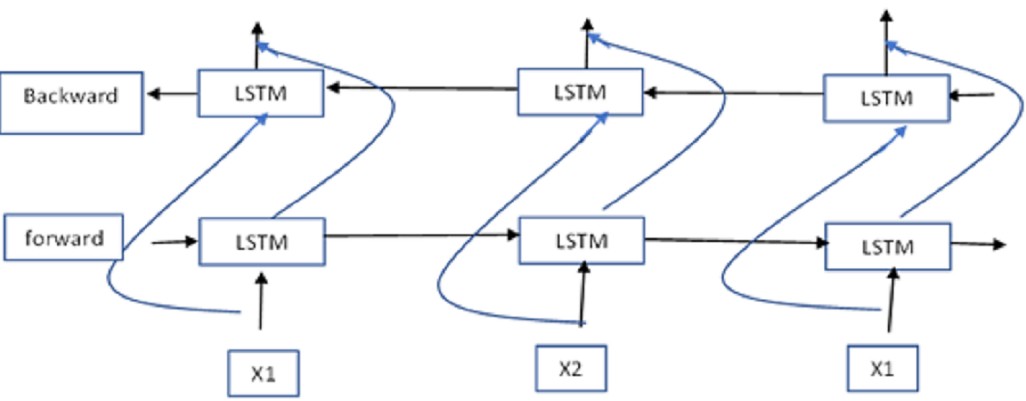

**Figure 4 The Bi-LSTM architecture.**

represented by a function $f$ such that $E_k = f(u_{k1}, u_{k2}, \ldots, u_{kl_n})$, where $E_k$ represents the embedding of the $k$th phrase.

## Attention module

The attention module aims to highlight the most crucial aspects of every given news item. Let us use a hypothetical five-sentence post to demonstrate this point. A deep network processes these sentences and generates a state variable $K$ at each stage. $K_1$ mostly covers the sentences $se_1$ and $se_2$ (and maybe some of $se_3$), whereas $K_5$ concentrates on $se_5$ and beyond. Attaining the attention weights $a_k$ ($k = 1, 2, \ldots, 5$) is the responsibility of the attention module, which is typically implemented as a simple two-layer neural network. After that, we build the post embedding $PD$ by averaging the states using a weighted formula:

$$PD = \sum_{k=1}^{5} a_k K_k. \tag{3}$$

To calculate $g(H_i, q)$, multiplicative attention makes advantage of inner product similarity, as shown below:

$$f(K_k, p) = \langle w^1 K_k, w^2 p \rangle. \tag{4}$$

Based on the task's objective function, BP is used to learn the weight matrices $w^1$ and $w^2$.

Each $K_k$ state in the input post undergoes a linear transformation and *tanh* activation in this procedure. It then performs an inner product of that number with the vector $p$ that serves as its context:

$$f(K_k, p) = \langle K_k, p \rangle, \tag{5}$$

where,

$$K_k = tan. \tag{6}$$

The similarity score between $H_i$ and $q$ is calculated using additive attention as follows:

$$f(K_k, p) = \omega^T \sigma(w^1 K_k + w^2 p), \tag{7}$$

where $\omega^T$ denotes the weight vector and the activation $tanh$ is denoted by $\sigma$.

## Customized loss function

Here, we focus on each class's mean squared error (MSE) to create a novel loss function. At the end of the network training phase, we used the validation data to determine whether a given model had improved upon the loss function's output.

$$E_{MSE} = \frac{1}{M} \sum_{k=1}^{M} (t_k - p_k)^2, \tag{8}$$

where $t_k$ and $p_k$ are the actual and the predicted values of the $k$th sample data from our dataset.

$$E_k = \frac{1}{N} \sum_{i=1}^{N} (t_{i,k} - p_{i,k})^2, \tag{9}$$

where $E_k$ denotes the loss value of the $k$th class.

$$L_d = \frac{1}{k} \sum_{i=1}^{k} (2a_i E_i)^2. \tag{10}$$

# EXPERIMENTAL SETUP

Python is used as the testing platform throughout all of the studies. The tests can only be carried out using the Python libraries known as Keras, NLTK, NumPy, Pandas, and Sklearn. We assess the presentation of the system based on many criteria, including accuracy, F-score, precision, and recall.

## Datasets

We have used the Weibo, liar, PHEME, fake and real news, and Buzzfeed datasets to analyze fake and real news.

### Weibo dataset

In the dataset compiled by *Jin et al. (2017)*, the genuine news articles were sourced from a reputable news outlet known as the Xinhua News Agency, while the fake news was verified using Weibo's official rumor debunking mechanism. This dataset was employed to assess the effectiveness of the introduced model. Our focus was exclusively on tweets that included both text and photos, allowing us to incorporate both textual features and visual elements. Consequently, tweets lacking either text or photos were removed. The data splitting technique employed aligns with the benchmark scheme, and data preprocessing follows a methodology similar to that outlined by *Jin et al. (2017)*. For a comprehensive breakdown of the dataset's statistical characteristics, refer to Fig. 5.

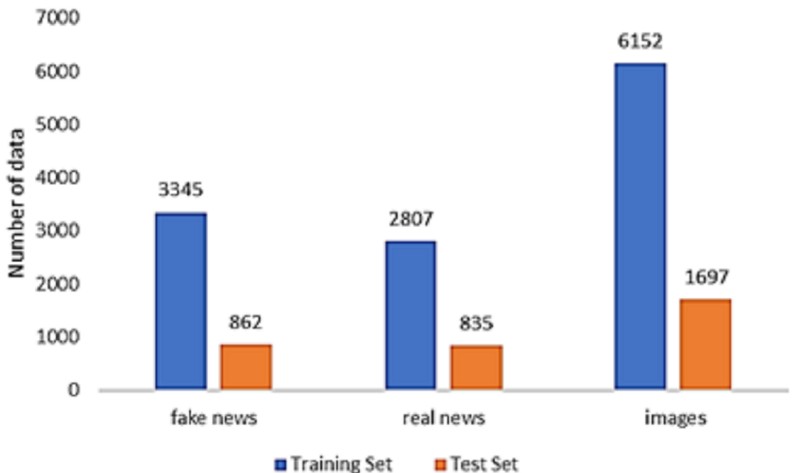

**Figure 5  Data distribution of the Weibo dataset.**

| | Unnamed: 0 | title | text | label |
|---|---|---|---|---|
| 0 | 8476 | You Can Smell Hillary's Fear | Daniel Greenfield, a Shillman Journalism Fello... | FAKE |
| 1 | 10294 | Watch The Exact Moment Paul Ryan Committed Pol... | Google Pinterest Digg Linkedin Reddit Stumbleu... | FAKE |
| 2 | 3608 | Kerry to go to Paris in gesture of sympathy | U.S. Secretary of State John F. Kerry said Mon... | REAL |
| 3 | 10142 | Bernie supporters on Twitter erupt in anger ag... | — Kaydee King (@KaydeeKing) November 9, 2016 T... | FAKE |
| 4 | 875 | The Battle of New York: Why This Primary Matters | It's primary day in New York and front-runners... | REAL |

**Figure 6  Collected sample data from the PHEME dataset.**

### PHEME dataset

The PHEME dataset comprises tweets, including rumors and non-rumors, posted on Twitter during significant breaking news events, as depicted in Fig. 6. To be more specific, it offers conversation threads from Twitter related to various major events, such as the Ferguson unrest, the Charlie Hebdo massacre, the Ottawa shooting, the Sydney hostage crisis, the Germanwings jet crash, and others.

The data structure is as follows: Within the directory, you'll find two folders labeled "rumors" and "non-rumors." Both of these folders contain subfolders named with a tweet ID, allowing you to locate the respective tweet by navigating the "source-tweet" directory. Conversely, the "reactions" directory contains a collection of tweets written in response to the source tweet. Moreover, each subfolder includes a file named "annotation.json" that provides information about the rumor's credibility and a file named "structure.json" that outlines the conversation's flow.

### Buzzfeed news dataset

The Buzzfeed news dataset is a comprehensive sample of news articles published on Facebook during the week leading up to the 2016 United States presidential election, specifically from September 19 to September 23, as well as September 26 and 27. These dates were chosen randomly. Each post and its associated story were individually reviewed

**Table 1 Primary characteristics of the Buzzfeed news dataset.**

| Parameter | Characteristics |
|---|---|
| Id | The "Id" that was allocated to the website for the news story. If the article is authentic, the status will be actual; otherwise, it will be phony |
| Title | This is a reference to the headline that is intended to grab the attention of readers and is relevant to the primary focus of the news story |
| Text | The "text" of the article, which expands on the news item. The publisher's perspective is shaped by the main claim, which is usually emphasized and elaborated on. |
| Source | It names a journalist who wrote the news piece or a publication outlet |
| Images | Pictures help readers understand a news story |
| Movies | A news article's video or movie clip link helps contextualize the story. Movies are crucial to the news. |

**Table 2 Fake and real news dataset.**

| Data type | Number of data |
|---|---|
| News | 9,050 |
| Politics | 6,841 |
| Left news | 4,459 |
| Government news | 1,570 |
| US news | 783 |
| Middle East news | 778 |

by five BuzzFeed editors to assess the accuracy of the claims made. There are two distinct datasets within the Buzzfeed news collection: one dataset contains false news, while the other comprises genuine news. Both datasets are presented in CSV format, with each containing 91 entries and 12 variables. The Buzzfeed news dataset is divided into two separate datasets, each featuring the primary characteristics listed in Table 1.

### Fake and real news dataset

In this study, we used a dataset compiled and made available to the public by *Ahmed, Traore & Saad (2018)*. The data from this dataset is illustrated in Table 2.

The sample data from this fake and real news dataset is shown in Fig. 7. This dataset consists of 23,481 data, which comprises news, politics, left news, government news, US news, and Middle East news.

## Data pre-processing

During this step, the provided datasets were pre-processed to remove the noise, which included things like stop words, punctuation marks, HTML tags, URLs, and emoticons, among other things. The NLTK toolkit, an open-source natural language processing package, was used for the pre-processing.

### Tokenization

Dividing text/string into tokens is the initial stage in natural language processing before feature extraction.

| | title | text | subject | date |
|---|---|---|---|---|
| 0 | Donald Trump Sends Out Embarrassing New Year'... | Donald Trump just couldn t wish all Americans ... | News | December 31, 2017 |
| 1 | Drunk Bragging Trump Staffer Started Russian ... | House Intelligence Committee Chairman Devin Nu... | News | December 31, 2017 |
| 2 | Sheriff David Clarke Becomes An Internet Joke... | On Friday, it was revealed that former Milwauk... | News | December 30, 2017 |
| 3 | Trump Is So Obsessed He Even Has Obama's Name... | On Christmas day, Donald Trump announced that ... | News | December 29, 2017 |
| 4 | Pope Francis Just Called Out Donald Trump Dur... | Pope Francis used his annual Christmas Day mes... | News | December 25, 2017 |
| ... | ... | ... | ... | ... |

**Figure 7 Sample data from fake and areal news datasets** *Ahmed, Traore & Saad (2018).*

### Word removal

Remove stop words after tokenizing. Stop words are minor words that produce noise in text categorization. These words help sentences organize and link words. Stop words include articles, prepositions, conjunctions, and pronouns.

### Stemming

Stemming reduces words to their roots (also known as lemma). Stemming reduces derivative words. The lemma of running ran, and the runner is run. The Porter stemmer algorithm, the most used stemming algorithm, was employed.

### Extraction

This research found 26 characteristics. Due to irrelevant features decreasing model accuracy and training cost, fewer features were chosen. Selecting several attributes also increases model training time. Thus, we decided on less effective measures like the number of words, characters, sentences, average word length, average sentence length, and name entity recognition-based features. For the named entity recognition feature, we retrieved person, org, date, time, facilities (airports, buildings, *etc.*), geopolitical entities (countries, cities, *etc.*), product, piece-of-art (book titles, music names, *etc.*), language, money, and cardinal from the text.

## RESULTS AND DISCUSSION

As a means of carrying out an analysis of the findings, we made use of four different metrics, all of which are predicated on the number of true positives (*TP*), false positives (*FP*), true negatives (*TN*), and false negatives (*FN*) in the predictions of the binary classifiers:

1. Accuracy, also known as the proportion of true forecasts (sometimes known as "right" predictions):

$$Accuracy(A) = \frac{TP + TN}{TP + TN + FP + FN}. \tag{11}$$

2. Recall, which measures the capability of the classifier to locate all of the positive samples in the data set:

$$Recall(R) = \frac{TP}{TP + FN}. \tag{12}$$

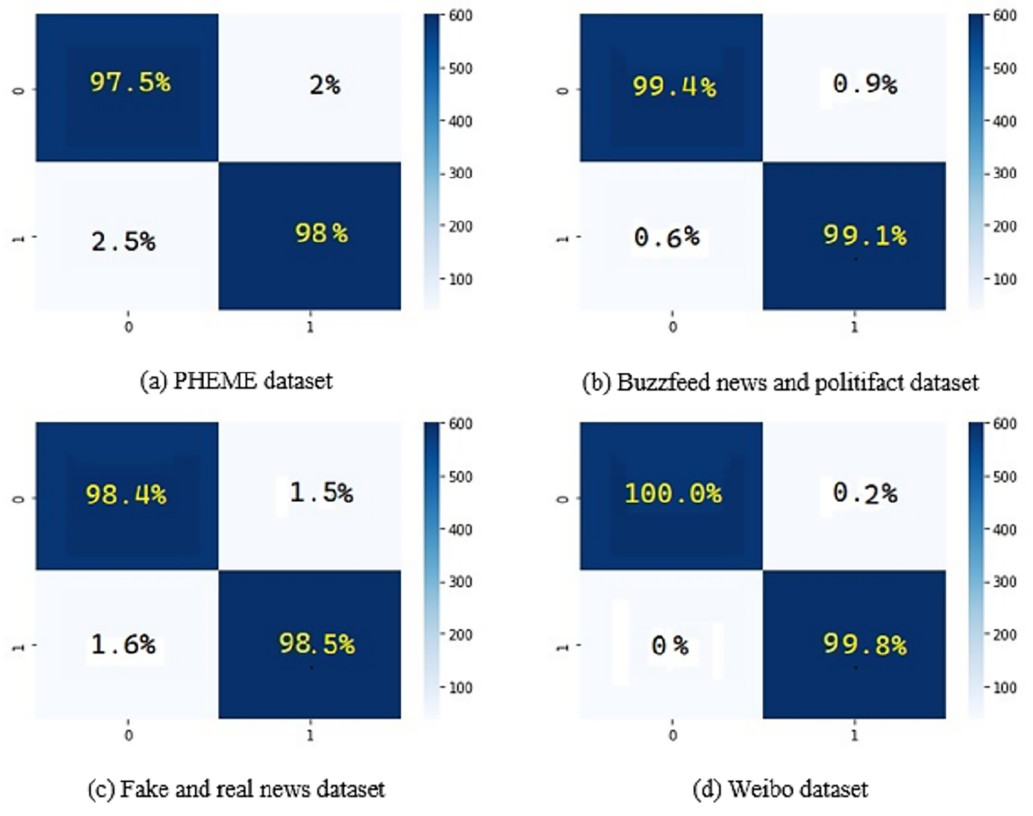

**Figure 8 Confusion matrix for the prediction of fake news on the datasets.**

3. Precision is determined as:

$$Precision(P) = \frac{TP}{TP + FP}.$$ (13)

4. The values that are computed for the F1-score, which is the harmonic mean of accuracy and recall, fall within the range [0, 1]:

$$Fl - Score(F1) = \frac{2 * (P * R)}{P + R}.$$ (14)

Figure 8 shows the confusion matrix on the different datasets using the proposed methodology DeepFND.

The statistical significance of the data was determined with the use of a paired t-test. The experiments were carried out five times (with five-fold cross-validation, meaning an 80–20% split each time), and the accuracy of the results was determined using 95% confidence intervals.

The results for the PHEME dataset using the proposed DeepFND are shown in Table 3. The overall accuracy of the proposed method is 98%.

**Table 3 Results on the PHEME dataset.**

|  | Precision | Recall | F1-score |
|---|---|---|---|
| Fake | 0.98 | 0.98 | 0.98 |
| Real | 0.97 | 0.98 | 0.98 |
| Accuracy | – | – | 0.98 |
| Macro avg | 0.97 | 0.98 | 0.97 |
| Weighted avg | 0.98 | 0.98 | 0.97 |

**Table 4 Results for the Buzzfeed news and politifact dataset.**

|  | Precision | Recall | F1-score |
|---|---|---|---|
| Fake | 0.98 | 1.00 | 0.99 |
| Real | 0.99 | 0.98 | 0.98 |
| Accuracy | – | – | 0.98 |
| Macro average | 0.97 | 0.98 | 0.97 |
| Weighted average | 0.97 | 0.98 | 0.97 |

Table 3 displays the performance results of a classification model on the PHEME dataset, which involves assessing the credibility of information as either "Fake" or "Real." The table presents metrics including precision (the accuracy of positive predictions), recall (the ability to identify positive instances), F1-score (the harmonic mean of precision and recall), and accuracy (overall correctness). The model demonstrates effective classification with high values of 0.98 for precision, recall, and F1-score for both "Fake" and "Real" classes, as well as an accuracy of 0.98. The macro and weighted averages further reinforce the model's accuracy, reflecting its ability to differentiate between classes while considering potential imbalances.

Table 4 showcases the performance outcomes of a classification model on the Buzzfeed news and Politifact dataset, designed to distinguish "Fake" from "Real" news. The table presents precision (the accuracy of positive predictions), recall (the ability to identify positive instances), F1-score (the harmonic mean of precision and recall), and accuracy (overall correctness). With high values of 0.98 and 1.00 for precision and recall, respectively, for the "Fake" class, and likewise impressive metrics for the "Real" class, including 0.99 precision and 0.98 recall, the model exhibits accurate classification with an overall accuracy of 0.98. Both macro and weighted averages confirm the model's efficacy in balanced performance assessment.

Moving to Table 5, we present the outcomes for the fake and real news datasets. The results underscore the model's effectiveness, with a precision of 0.99 and recall of 1.00 for the "Fake" class, along with perfect scores for the "Real" class. This leads to an impressive accuracy of 0.99, reflecting the model's accurate classification. The macro and weighted averages reinforce these outcomes, demonstrating consistent and precise performance across both classes.

**Table 5 Results for the fake and real news dataset.**

|  | Precision | Recall | F1-score |
| --- | --- | --- | --- |
| Fake | 0.99 | 1.00 | 0.99 |
| Real | 1.00 | 0.99 | 0.99 |
| Accuracy | – | – | 0.99 |
| Macro average | 0.99 | 0.99 | 0.99 |
| Weighted average | 0.99 | 0.99 | 0.99 |

**Table 6 Results for the Weibo data set.**

|  | Precision | Recall | F1-score |
| --- | --- | --- | --- |
| Fake | 1.00 | 1.00 | 1.00 |
| Real | 1.00 | 0.99 | 0.99 |
| Accuracy | – | – | 0.99 |
| Macro average | 1.00 | 1.00 | 0.99 |
| Weighted average | 1.00 | 0.99 | 0.99 |

Finally, in Table 6, we provide details of the model's performance on the Weibo dataset. Exceptional scores of 1.00 for precision and recall in the "Fake" class, along with highly commendable scores in the "Real" class, result in an accuracy of 0.99. The macro average upholds the model's high performance, while the weighted average aligns with the macro results, underscoring its consistency in effective classification.

The results in the case of the Buzzfeed news and Politifact dataset are shown in Table 4. The overall accuracy for this dataset is achieved as 98.43%.

The results for the fake and real news datasets are shown in Table 5.

The results for the Weibo dataset are shown in Table 6.

Evaluating our algorithms using cross-validation is not the most illuminating method available. When determining whether a news item is a hoax, there is a cost associated with creating the training set. This is because each post may need to be examined individually. The more intriguing issue is not how accurate of a level we can achieve when we know the ground truth for 80% of the postings; instead, the more relevant question is how large of a training set we need to reach a specific degree of accuracy. Our methods need to create an accurate classification while depending on a small portion of posts already known to belong to a particular category for us to scale up to the extent of the information sharing that occurs inside social networks.

For the fold-2 phase, the values of our proposed model DeepDND's training loss and training accuracy are shown in Figs. 9 and 10, respectively. When the loss values for the multi-class classification were analyzed, the model showed a rapid decline in the loss value, and it got very close to the zero value (Fig. 9). On the other hand, it was found that our model had a more practical procedure for learning new information.

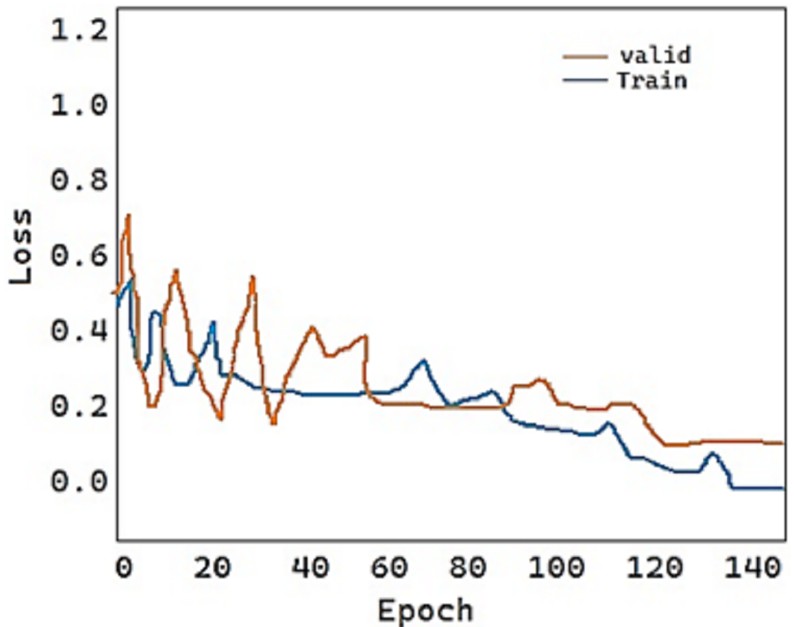

**Figure 9 Train and validation loss of the proposed model.**

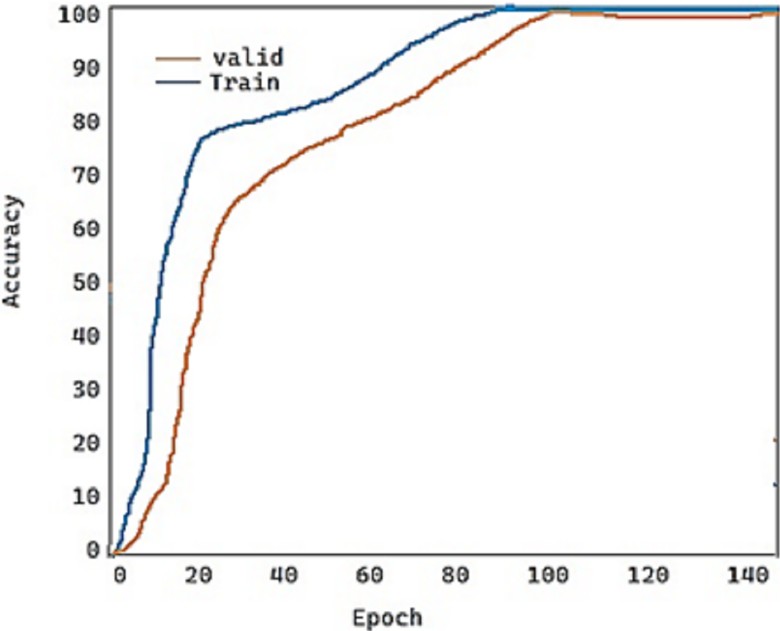

**Figure 10 Train and validation accuracy of the proposed model.**

All the approaches' receiver operating characteristic (ROC) curves on both datasets are also displayed for thoroughness' sake. The true positive rate (TPR) and the false positive rate (FPR) are plotted to produce the ROC curve, which is then used to diagnose a binary

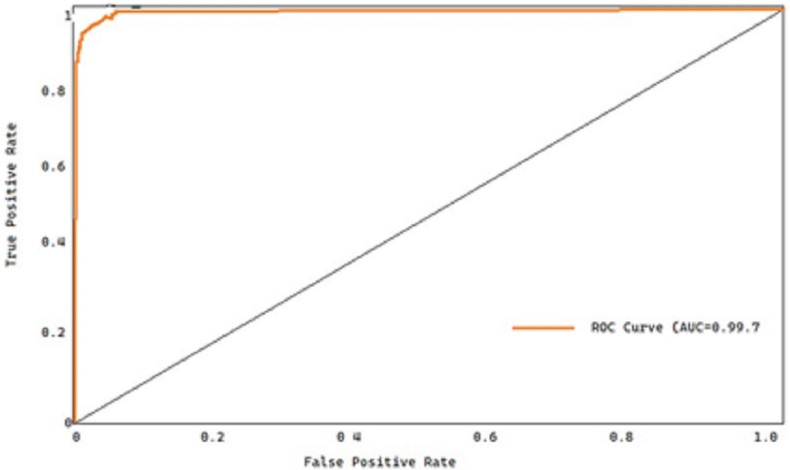

**Figure 11** ROC curve of the proposed model.

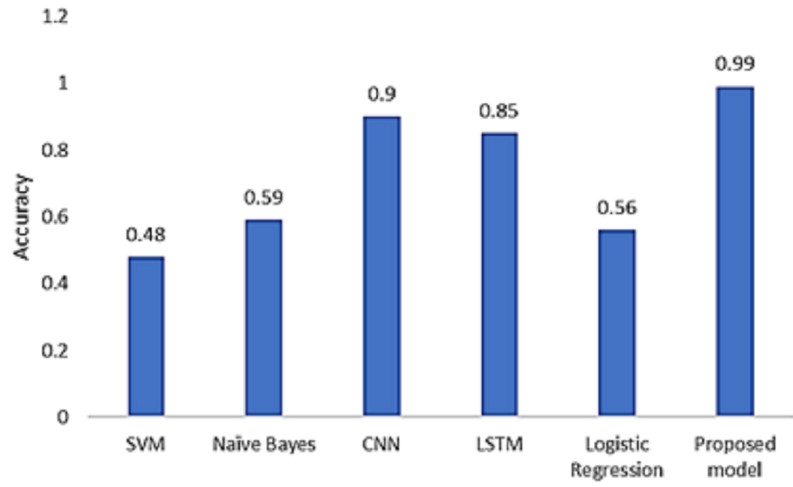

**Figure 12** Model comparison with existing models.

classifier. Figure 11 shows the receiver operating characteristic curves for the overall dataset.

We compared our technique with the baseline models. According to the results depicted in Fig. 12, our model produces the highest accuracy in classifying real and fake news.

## CONCLUSION AND FUTURE WORK

The realm of deception detection within social media is a burgeoning domain of inquiry, and researchers are actively grappling with the intricacies of devising efficacious approaches amidst the exponential proliferation of the spurious news landscape. This research promises to offer pivotal insights that could guide forthcoming inquiries into crafting optimal methodological amalgamations aimed at unmasking insincere content pervading social media platforms.

A more precise false news detection algorithm is the goal of the approach presented in this research. A new fine-grained ensemble network called DeepFND is introduced. It fully fuses textual characteristics with visual data to detect false news. DeepFND utilizes a VGG-19 and Bi-LSTM ensemble model to combat the spread of incorrect information on social media. This system extracts features from the article's text and images using an ensemble DL approach. The ensemble employs the joint feature extractor and attention modules using pre-training and fine-tuning phases. A novel tailored loss function is utilized in the current study. Automatic approaches for identifying online fake news are investigated. The fusion is fine-grained and sufficient as it considers the interdependencies between various visual characteristics and textual data. DeepFND is found to be effective by experiments conducted on publicly available datasets. Compared to other approaches for merging visual and textual representations, DeepFND performs well. The DeepFND's joint model, which combines optical and linguistic characteristics, is demonstrated to outperform the collective term created by fusing visual and textual representations.

Future work will include incorporating textual and visual information based on social context. The frequency domain visual characteristics will also be evaluated to boost the efficiency of false news identification further.

### Funding
Princess Nourah bint Abdulrahman University Researchers Supporting Project number (PNURSP2022R263), Princess Nourah bint Abdulrahman University, Riyadh, Saudi Arabia supported the APC for this article. The funders had no role in study design, data collection and analysis, decision to publish, or preparation of the manuscript.

### Grant Disclosures
The following grant information was disclosed by the authors:
Princess Nourah bint Abdulrahman University: PNURSP2022R263.
Princess Nourah bint Abdulrahman University, Riyadh, Saudi Arabia.

### Competing Interests
Vladimir Simic is an Academic Editor for PeerJ.

### Author Contributions
- Venkatachalam K conceived and designed the experiments, analyzed the data, authored or reviewed drafts of the article, and approved the final draft.
- Badriyya B. Al-onazi conceived and designed the experiments, performed the computation work, prepared figures and/or tables, and approved the final draft.
- Vladimir Simic conceived and designed the experiments, performed the experiments, analyzed the data, performed the computation work, prepared figures and/or tables, authored or reviewed drafts of the article, and approved the final draft.
- Erfan Babaee Tirkolaee performed the experiments, performed the computation work, authored or reviewed drafts of the article, and approved the final draft.

- Chiranjibe Jana performed the experiments, authored or reviewed drafts of the article, and approved the final draft.

## Data Availability

The data are available at Weibo-COV: https://paperswithcode.com/dataset/weibo-cov.

The code is available at Kaggle: https://www.kaggle.com/datasets/usharengaraju/pheme-dataset.

## Supplemental Information

Supplemental information for this article can be found online at http://dx.doi.org/10.7717/peerj-cs.1666#supplemental-information.

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
