# Peer review of "DeepFND: an ensemble-based deep learning approach for the optimization and improvement of fake news detection in digital platform"

_PeerJ Computer Science, doi:10.7717/peerj-cs.1666_

## Round 0.1 · original submission · Major Revisions

Dear authors,

Thank you for your submission. Your article has not been recommended for publication in its current form. However, we do encourage you to address the concerns, issues, and criticisms of the reviewers and resubmit your article once you have updated it accordingly.

Best wishes,

**Language Note:** PeerJ staff have identified that the English language needs to be improved. When you prepare your next revision, please either (i) have a colleague who is proficient in English and familiar with the subject matter review your manuscript, or (ii) contact a professional editing service to review your manuscript. PeerJ can provide language editing services - you can contact us at copyediting@peerj.com for pricing (be sure to provide your manuscript number and title). – PeerJ Staff

Reviewer 1 ·

Basic reporting

no comment

Experimental design

The following points need to be addressed:
1. The abstract you have mentioned the ensemble deep learning model. you might include the techniques in deep. You must consist of the novelty of the research in abstract.
2. In the introduction give the list of motivations for research using deep learning models. Also, contributions must be discussed in detail.
3. Introduction section could be shortened. Vague content can be eliminated
4. The introduction does not have the problem definition and scope of the research in detail.
5. proposed model does not clearly state the word embedding and feature extraction in detail. Also, text analysis process is missing.
6. It needs to be clarified what are deep learning models used in the ensemble techniques. The all the deep learning models need to be explained in the proposed model. Also, describe the model’s efficiency in detail.
7. This paper has the required quality. It has described the literature review, existing problem and proposed work well.
8. DEEPFND is not discussed are defined in the proposed section.

Validity of the findings

The following points need to be addressed:
1.The result section needs to be discussed in detail. The text analysis result needs to be stated in the result section.
2.Conclusion must discuss the achievements of the novel method and research objective in detail.

Reviewer 2 ·

Basic reporting

There is no mathematical model discussed in the deep learning section. The author must specify the mathematical model used in text analysis

The workflow diagram is not stated in the proposed section.

Provide more context on the challenges faced by fake text analysis. It would be beneficial to provide more specific details about the difficulties in managing data and semantics in the study due to the explosive expansion of the social media system.

Additionally, citing relevant statistics or industry trends could further support the importance of the research.

Experimental design

Clearly define the proposed ensemble deep learning technique. While the abstract mentions the deep learning technique for fake data prediction, the procedure in the proposed section lacks specific details on how the technique combines different algorithms or methods.

Providing a clear explanation of the hybrid approach will help readers understand the novelty and potential advantages of the proposed method.

Elaborate on the Attention Neural Network for text analysis: Including bandwidth and resource load in the NN for effective resource allocation, but it lacks details on achieving this.

Expanding on the NN architecture and how it addresses design limitations could give readers a better understanding of the proposed solution.

Validity of the findings

The major advantages of the proposed methodology need to be highlighted in the manuscript. The results sections could be still improved and proper justification is required for each graph.

---

## Round 0.2 · accepted · Accept

Dear authors,

Thank you for the revision. The paper seems to be improved in the opinion of the reviewers. The paper is now ready to be published.

Best wishes,

Reviewer 1 ·

Basic reporting

no comment

Experimental design

no comment

Validity of the findings

no comment

Additional comments

The author have addressed all the points given for revision.
Adequate work done by the author.

Reviewer 2 ·

Basic reporting

All reviewer questions were addressed.

Experimental design

Satisfied

Validity of the findings

Valid